# Investigation of Lung Cancer Cell Response to Cryoablation and Adjunctive Gemcitabine-Based Cryo-Chemotherapy Using the A549 Cell Line

**DOI:** 10.3390/biomedicines12061239

**Published:** 2024-06-03

**Authors:** Kimberly L. Santucci, Kristi K. Snyder, Robert G. Van Buskirk, John G. Baust, John M. Baust

**Affiliations:** 1CPSI Biotech, Owego, NY 13827, USA; 2Phase Therapeutics, Inc., Owego, NY 13827, USA; 3Center for Translational Stem Cell and Tissue Engineering, Binghamton University, Binghamton, NY 13902, USA; 4Department of Biological Sciences, Binghamton University, Binghamton, NY 13902, USA

**Keywords:** cryoablation, lung cancer, NSCLC, chemotherapy, necroptosis, apoptosis, gemcitabine, combination treatment, bronchoscopy, NOTES

## Abstract

Due to the rising annual incidence of lung cancer (LC), new treatment strategies are needed. While various options exist, many, if not all, remain suboptimal. Several studies have shown cryoablation to be a promising approach. Yet, a lack of basic information pertaining to LC response to freezing and requirement for percutaneous access has limited clinical use. In this study, we investigated the A549 lung carcinoma cell line response to freezing. The data show that a single 5 min freeze to −15 °C did not affect cell viability, whereas −20 °C and −25 °C result in a significant reduction in viability 1 day post freeze to <10%. These populations, however, were able to recover in culture. Application of a repeat (double) freeze resulted in complete cell death at −25 °C. Studies investigating the impact of adjunctive gemcitabine (75 nM) pretreatment in combination with freezing were then conducted. Exposure to gemcitabine alone resulted in minimal cell death. The combination of gemcitabine pretreatment and a −20 °C single freeze as well as combination treatment with a −15 °C repeat freeze both resulted in complete cell death. This suggests that gemcitabine pretreatment may be synergistically effective when combined with freezing. Studies into the modes of cell death associated with the increased cell death revealed the increased involvement of necroptosis in combination treatment. In summary, these results suggest that repeat freezing to −20 °C to −25 °C results in a high degree of LC destruction. Further, the data suggest that the combination of gemcitabine pretreatment and freezing resulted in a shift of the minimum lethal temperature for LC from −25 °C to −15 °C. These findings, in combination with previous reports, suggest that cryoablation alone or in combination with chemotherapy may provide an improved path for the treatment of LC.

## 1. Introduction

Lung cancer (LC) is the second most common cancer in both men and women behind skin cancer, with an estimated 238,340 new cases diagnosed in 2023 [1,2]. Of these, approximately 10–15% are small-cell lung cancer (SCLC), while the remaining 85% are non-small-cell lung cancer (NSCLC) [3]. The American Cancer Society reports about 1 in 5 cancer-related deaths to be caused by LC, resulting in more deaths than colon, breast, and prostate cancer combined [1]. These statistics reveal the urgent need for improved treatment options. Treatment for NSCLC is dependent on the stage and includes surgery, radiation therapy, chemotherapy, targeted drug therapy, and immunotherapy, often in combination [4,5]. NSCLC that has spread is more difficult to treat, but an increasing number of targeted therapies and immunotherapy drugs are used, depending on the cancer’s genetic profile and biomarkers.

Cryoablation (CA) is an established treatment modality for a variety of cancers and has been shown to be superior to radiation therapy in the treatment of NSCLC in a recent meta-analysis [6]. Additionally, CA has been used for a number of years for medically inoperable NSCLC, as well as pulmonary metastases, with encouraging results [7,8,9,10,11,12]. Traditional CA devices utilize compressed gases, such as argon, in combination with percutaneous cryoprobes to apply ultra-cold temperatures to targeted tissue [13]. The necessity of a percutaneous approach has limited the use of CA for LC due to associated complications including high risk of pneumothorax [14,15]. When CA is utilized, freeze duration, cryoprobe size, cryogen choice, and number of freeze/thaw cycles offer a variety of technique modifications to suit various tumor sizes and locations [16]. The lethal temperature varies between cancer and tissue types, necessitating careful planning and visualization (tumor and frozen mass) to achieve maximum targeted destruction while avoiding damage to adjacent structures [17,18]. For instance, studies have shown that for the destruction of pancreatic and renal cancer, a minimal lethal temperature (MLT) of <−25 °C must be attained, whereas for late-stage prostate cancer, temperatures <−40 °C are necessary [19,20,21,22,23]. Application of a double freeze (providing repeat thermal insult in a finite window) has been shown to increase the MLT in the range of 5 to 10 °C [19,20,21,22,23]. In the case of LC, while numerous studies have demonstrated that CA can be used to effectively target NSCLC and support lethality in the −20 °C to −35 °C range [24,25,26], the definition of the MLT for LC remains unclear. Defining the ablative region within a cryolesion can be complicated as the primary modes and mechanisms of cell death vary following exposure to different temperatures [17,27]. For instance, necrotic tissue dominates the core of the lesion where temperatures are typically <−40 °C, while in the periphery of a cryolesion where temperatures range between −40 °C and 0 °C (nominally) at the edge of the ice ball, there is a mixed population of necrotic, apoptotic, and live cells. It is within this region wherein there is a transition zone from predominantly dead/dying (necrotic and apoptotic) cells to cells which survive the freeze/thaw episode. This transition zone can be correlated directly to a temperature, thereby representing the MLT. While referred to as the MLT, in practice it is important to recognize that this represents a minimal target. Understanding the MLT can have significant impacts on procedural outcome by increasing the probability of cancer destruction while enabling the reduction of unnecessary damage to surrounding non-targeted tissue (extent of collateral damage).

In addition to the efficacy of CA alone, the use of adjuvants to increase the lethality of freezing has been well documented in many cancer types, including prostate [28,29,30,31], kidney [32,33,34], breast [35,36,37], lung [38,39,40,41], among others [17,19,42,43,44]. Adjuvants can include traditional chemotherapeutic agents (drugs), other thermal ablation strategies, small molecules or immunotherapies, to name a few [17,38,42,45,46]. These, and numerous other reports, have demonstrated that the combination of CA with adjunctive chemotherapy results in a shift in the susceptibility of cancer cells to sublethal temperatures and, as such, a shift in the MLT to a warmer temperature [17,42,43,47]. To this end, gemcitabine is an antimetabolite chemotherapeutic agent utilized in the treatment of all forms of NSCLC including adenocarcinoma [48,49,50,51]. Gemcitabine is on the American Cancer Society’s list of Most Commonly Used Chemotherapy Agents for NSCLC [52]. A PubMed search using the keywords “gemcitabine and lung adenocarcinoma and clinical study” yields >4600 publications. In addition to use as a first- and second-line treatment of lung adenocarcinoma, gemcitabine is also used as an adjuvant either before or after surgery, or in combination with other treatments (immunotherapy, thermal ablation) for patients who are not good surgical candidates [53,54,55,56]. Its efficacy in NSCLC has been noted since the early 1990s [57], and it is generally well tolerated. Based upon the promising initial clinical reports on the use of cryoablation to treat NSCLC as well as the potential of the use of freezing in combination with standard-of-care chemotherapy, in this study we conducted a series of in vitro investigations in an effort to establish the MLT of NSCLC using the A549 cell line, as well as the potential of combinatorial gemcitabine/freezing treatment to increase A549 cell death. A549 cells were selected as the model cell system as they are a well-established and commonly utilized adenocarcinoma cell model for in vitro NSCLC studies.

## 2. Materials and Methods

### 2.1. Cell Culture

The NSCLC A549 lung adenocarcinoma cell line (CCL-185, ATCC) was cultured in T-75 flasks (Cell Treat, Shirley, MA, USA) with F12K medium (ATCC) containing 10% Fetal Bovine Serum (FBS, Peak Serum, Bradenton, FL, USA) and 1% penicillin/streptomycin (Lonza, Walksersville, MD, UA). Cells were plated at 1.6 x 10^3^ cells/cm^2^ in Costar 96-well strip-well plates or 35 mm dishes and grown for 24 h prior to experimentation.

### 2.2. Gemcitibine Exposure

Gemcitabine (Enzo Life Sciences, Farmingdale, NY, USA) working stock was prepared fresh in sterile water (VWR, Radnor, PA, USA) prior to sample application and diluted to final concentrations in culture media. Samples were then treated with varying concentrations of gemcitabine for 24 h. Following the exposure interval, medium containing gemcitabine was removed, and fresh medium was applied 30 min prior to freezing (combination studies) or returned to culture (dose–response studies).

### 2.3. Freeze Procedure

Samples in Costar strip wells were frozen to final temperatures of −10 °C, −15 °C, −20 °C, and −25 °C using a 5 min single or repeat (double) freeze protocol in an aluminum sample freezing block fixture within a refrigerated circulating bath (Neslab/Thermo, Waltham, MA, USA). The freezing block fixture contained a thin coating of ethanol to facilitate uniform thermal contact and heat exchange between sample wells within the block. Sample temperature was monitored with a type-T thermocouple, and ice nucleation was initiated using liquid nitrogen vapor at −2 °C to prevent supercooling. Samples were cooled for a total of 5 min and had a minimum hold time of 30 s at the designated nadir target temperature, then thawed passively for 10 min under a laminar flow hood. For repeat exposure (double freeze), following thawing, samples were subjected to a second freeze/thaw protocol. Following completion of the single or repeat freeze/thaw protocol, samples were placed into a 37 °C incubator for recovery and downstream assessment.

### 2.4. Cell Viability Assessment

Metabolic activity was assessed over 7 days post treatment using alamarBlue (Invitrogen/Molecular Probes, Waltham, MA, USA) diluted 1:20 in HBSS (Corning, Tewksbury, MA, USA). Briefly, culture medium was removed, and diluted alamarBlue was applied for 30 min (±1 min) at 37 °C. Fluorescence was measured using a Tecan Infinite plate reader with excitation 530 nm and emission 590 nm (Tecan Austria GmBH, Grodig, Austria). Background-subtracted raw fluorescent units were converted to percentage of pre-freeze control viability values and analyzed using Microsoft Excel.

### 2.5. Flow Cytometry Assessment of the Modes of Cell Death

CellEvent Green (ThermoFisher, Waltham, MA, USA, C10423) and SytoxRed (ThermoFisher, S11380) were used to assess apoptosis and necrosis, respectively, in samples following experimentation. Briefly, culture media and lifted cells were removed from sample wells and collected in individual 15 mL centrifuge tubes for each condition. Wells were rinsed with HBSS which was added to each respective tube. TrypLE express (Gibco/Invitrogen, Waltham, MA, USA) was then added to each sample well and incubated at 37 °C for 8 min to detach any adherent cells. Culture medium was then added to neutralize TrypLE, and the detached cell population was added to their respective 15 mL tube (pooled population). Samples were centrifuged at 300× *g* for 5 min, supernatant decanted, the resultant pellets resuspended in 500 uL 1x PBS containing CellEvent Caspase3/7 green, then samples were incubated at 37 °C for 25 min. SytoxRed was then added to each sample and incubated for an additional 5 min at 37 °C. Samples were transferred to a 96-well round-bottom plate in triplicate for flow cytometry analysis using a Guava EasyCyte Plus microcapillary system (Guava/EMD Millipore, Burlington, MA, USA and ExpressPro Software v5.3. Per manufacturer’s recommendations, sample concentrations were under 200 cells/µL (range: 75–200), and 5000 events per sample were acquired using a flow rate of 0.59 µL/s. Standard compensation techniques were tested using single-channel controls but ultimately unnecessary given the resolution of these fluorochromes. Gating was performed to exclude debris and doublets from analysis, and a quad stat marker was used to distinguish between live, necrotic, and apoptotic populations based on positive and negative single-channel controls. The results of three separate experiments were combined and analyzed using Microsoft Excel 365 (v2404).

### 2.6. Protien Immunoblot Analysis

Total protein was extracted from samples using radioimmunoprecipitation assay (RIPA) buffer containing protease and phosphatase inhibitors (HALT cocktail, ThermoFisher) and quantified using a BCA protein assay (Pierce/ThermoFisher, Waltham, MA, USA). Samples were then assessed using a WES Simple Western automated microcapillary system (Protein Simple/BioTechne, Minneapolis, MD, USA, 12–230 kDa separation kit) according to the manufacturer’s protocol. Lysates were diluted to 2 ug/uL, combined with the provided fluorescent master mix, denatured by heating to 95 °C for 5 min, then placed onto ice, and all subsequent steps were carried out on ice. Samples, antibody diluent, primary and secondary antibodies, luminol and peroxide, and wash buffers were aliquoted to appropriate locations within the plate, and then the plate was centrifuged at 1000× *g* for 5 min. Following centrifugation, the plate was inserted into the WES system along with a 25 capillary cartridge, and the run was initiated. Samples were separated for 28 min at 375 v, incubated with primary antibody for 60 min and secondary antibody for 30 min prior to detection. Antibodies of interest, MLKL (CST#14993, 1:50); Phospho-MLKL (CST#91689, 1:20); RIP3 (CST#13526, 1:20); and Phospho-RIP3 (CST#93654, 1:20), were multiplexed with vinculin (CST#4650, 1:100) for normalization and quantification. Data were collected using the Protein Simple’s Compass software (v 6.2.0) and were exported and analyzed using Microsoft Excel.

### 2.7. Data Analysis

Viability assessment was conducted on a minimum of three experimental repeats with an intra-experimental replicate of 7 (N ≥ 3; n ≥ 21) for each condition. Flow Cytometry analysis was conducted on a minimum of three experimental repeats with an intraexperimental replicate of 3 (N ≥ 3; n ≥ 9). Immunoblotting was conducted on a minimum of three experimental repeats. Within each individual experiment, protein samples were collected from a minimum of three replicates, pooled, and then analyzed (N ≥ 3; n ≥ 9). Data for all studies were analyzed using Microsoft Excel, and statistical analysis was conducted using single-factor ANOVA and Tukey-HDS post hoc test to determine statistical significance (astatsa.com/OneWay_Anova_with_TukeyHSD). *p* < 0.05 was regarded as statistically significant.

## 3. Results

### 3.1. Establishment of the Minimal Lethal Temperature for A549 Cells

A549 cells were exposed to −10, −15, −20, and −25 °C in either a single or repeat 5 min freeze protocol (Figure 1A and 1B, respectively). A single 5 min freeze to −10 °C or −15 °C yielded minimal cell death at 24 h post freeze (−10 °C: 164.5% (±1.2) and −15 °C: 96.4% (±2.8), versus control 137.9% (±0.8), respectively). A single freeze to −20 °C resulted in a significant decrease in 24 h viability to 10.6% (± 0.5) (*p* < 0.01). Samples, however, recovered to that of controls over the 7-day assessment period (D7: 122.2% (±2.4)). A single freeze to −25 °C yielded near-complete A549 cell destruction, resulting in a 24 h viability of 2.2% (±0.1). However, in vitro, the surviving cells were able to recover to 35.5% (±2.0) by day 7.

The impact of application of repeat (double) freeze protocol was then explored (Figure 1B). Similar to the single freeze, application of a repeat freeze at −10 °C resulted in minimal cell death (D1: 159.6% (±1.2)). When samples were treated with a repeat freeze to −15 °C, a significant decline in cell viability was observed at 24 h post treatment compared to −15 °C single freeze samples (D1: repeat −15 °C: 22.7% (±1.2) vs. single −15 °C: 96.4% (±2.8), *p* < 0.01); however, the surviving cells were able to recover in vitro over the 7-day assessment period (D7: 132.9% (±3.4)). Application of a repeat freeze to −20 °C resulted in 24 h survival of 1.6% (±0.1), but again the surviving cells were able to recover in vitro to 22.1% (±1.9) by day 7. When samples were exposed to a repeat freeze to −25 °C, complete cell death without any significant repopulation over the 7-day assessment period was observed (D1 = 0.4% (±0.1) vs. D7 = 1.7 (±0.1) (*p* = 0.06)).

### 3.2. Establishment of the Gemcitabine LC50 for A549 Cells In Vitro

A range of gemcitabine concentrations were applied to samples for 24 h, and then viability was subsequently assessed for 7 days to determine the LC50 (lethal concentration providing 50% cell death) for A549 cells in vitro (Figure 2). The growth characteristics of treated samples were compared to untreated control cells. The 10 nM and 25 nM treated samples yielded similar results, with viability of 94% (±2.2) and 93.6% (±2.4) upon drug removal followed by observed cell regrowth over the 7 day assessment period (149.1% (±1.0) and 150.8% (±1.0), respectively). Exposure to 50 nM of gemcitabine yielded 93.5% (±1.9) viability following drug removal (day 0) and a slight decrease in cell growth over the 7-day assessment interval (118.6% (±2.2)) (*p* < 0.01). Exposure to 75 nM and 100 nM of gemcitabine resulted in a substantial decrease in sample viability (increase in cell death) over the 7-day assessment interval, yielding 56.7% (±1.7) and 41.6% (±1.5) by day 7, respectively. As such, 75 nM was selected for combination studies.

### 3.3. Impact of Combination Gemcitabine Pretreatment and Freezing

With the establishment of 75 nM of gemcitabine resulting in ~50% cell death over the 7-day assessment period, this dose was selected for combination studies. Samples were exposed to 75 nM of gemcitabine for 24 h, then frozen to −15 °C, −20 °C, or −25 °C with a single or repeat freeze protocol, passively thawed, returned to culture, and viability was assessed for 7 days (Figure 3A,B). Pretreatment with 75 nM of gemcitabine followed by a single −15 °C freeze resulted in a significant decrease in viability compared to both −15 °C or gemcitabine alone (16.9% (±0.6) vs. 22.7% (±1.2) or 93.3% (±1.4), respectively, *p* < 0.01) (Figure 3A). Additionally, A549 regrowth was reduced in comparison to either −15 °C or gemcitabine alone samples (12.0% (±0.6) vs. 132.9% (±3.4) or 56.7% (±1.7) on day 7, respectively, *p* < 0.01). The combination of 75 nM of gemcitabine pretreatment and a single freeze to −20 °C or −25 °C resulted in complete cell death at 24 h (0.96% (±0.1) and 0.26% (±0.03), respectively, with no recovery at day 7 (0.6% (±0.05) and 0.2% (±0.02), respectively) compared to −20 °C or −25 °C freeze alone samples which recovered to 132.9% (±3.4) and 22.1% (±1.9), respectively, as well as 56.7% (±1.7) in gemcitabine alone samples.

When the combination protocol included a repeat freeze, in samples exposed to a repeat −15 °C freeze (Figure 3B), near-complete cell death was observed. Specifically, the gemcitabine/repeat −15 °C combination yielded 3.7% (±0.4) survival on day 1, which declined to 2.3% (±0.2) by day 7. This was compared to repeat −15 °C alone, which yielded 22.7% (±1.2) viability on day 1 and recovered to 132.9% (±3.4) by day 7 (*p* < 0.01 for combination vs. freeze alone at D1 and D7). As with the combination single freeze at −20 °C, application of a repeat freezing to −20 °C in combination with gemcitabine pretreatment yielded complete cell death with no recovery (D1 = 0.2% (±0.0) and D7 = 0.1% (±0.0)).

### 3.4. Assessment of the Mode of Cell Death Activated following Treatment

With the significant increase in cell death observed following the combination of gemcitabine pretreatment and freezing to −15 °C, microcapillary flow cytometry was utilized to assess the modes of cell death associated with the various treatments at 24 h post treatment (Figure 4 and Table 1). The percentage of live, necrotic, and apoptotic cells within a given population were determined by staining with Cell Event Green (apoptosis (activated caspase 3/7)) and Sytox Red (necrosis). Samples were gated to exclude cellular debris. Dot plots from one representative experiment are shown in Figure 4, and the average of the replicate experiments is presented in Table 1. Control samples yielded an average of 89.7% (±4.9) live, 7.9% (±5.2) necrotic, and 2.1% (±1.6) apoptotic cells. Samples treated with 75 nM of gemcitabine for 24 h resulted in a population distribution of 71.8% (±6.1) live, 19.9% (±7.3) necrotic, and 6.9% (±5.4) apoptotic cells. Samples exposed to a single −15 °C freeze yielded 64.1% (±6.6) live, 30.6% (±8.4) necrotic, and 4.0% (±2.9) apoptotic subpopulations. When the combination of gemcitabine pretreatment and a single −15 °C freeze was applied, a significant increase in the necrotic cell population with a corresponding decrease in live cells was observed. Specifically, the Gem/−15 °C samples yielded 27.7% (±2.5) live, 67.65% (±3.8) necrotic, and 2.9% (±1.0) apoptotic cell subpopulations.

Based on the observed increase in necrotic levels in combination-treated samples with little change in apoptotic levels, protein analysis was conducted to determine if the activation of necroptosis contributed to this increase. Accordingly, analysis of mixed-lineage kinase domain like pseudokinase (MLKL) and phospho-MLKL protein levels was conducted at 6 h post freeze comparing −15 °C freeze alone, 75 nM of gemcitabine (Gem) alone, Gem/−15 °C combination, and non-frozen 37 °C controls. MLKL protein levels were not found to differ significantly between any of the conditions (Figure 5A). Phospho-MLKL levels did not differ significantly between controls and gemcitabine-treated samples (*p* = 0.7). However, comparison of −15 °C freeze alone revealed a doubling in phospho-MLKL levels over controls (*p* = 0.04). Interestingly, when samples were treated with the combination of gemcitabine pretreatment and −15 °C freezing, a fourfold increase in phospho-MLKL levels compared to controls (*p* < 0.01) or gemcitabine alone (*p* < 0.01) samples was observed. Combination-treated sample phospho-MLKL levels were also found to be twofold greater than in −15 °C alone samples (*p* < 0.01, Figure 5B).

Analysis of receptor-interacting protein kinase 3 (RIP3 kinase), the upstream regulator of MLKL phosphorylation, was also conducted. RIP3 kinase levels were found to decrease, and phosphorylated RIP3K levels increased in freeze and combination samples compared to controls (*p* = 0.03 and *p* < 0.05, respectively), further supporting the involvement of necroptosis-mediated cell death (Figure 5C).

In addition to assessment of the necroptosis proteins, investigations were conducted to determine if pyroptosis and/or apoptosis were also involved in the early stages of cell death post treatment. Assessment of the protein Gasdermin D revealed no cleavage products, suggesting that pyroptosis was unlikely to be involved (Figure 6). Analysis of Caspase 3 revealed the presence of minimal cleavage products, suggesting lesser apoptotic involvement at 6 h post freeze (Figure 7). Taken together, these results suggested a primary role for necroptotic cell death in the combination of gemcitabine pretreatment and freezing in A549 cells.

## 4. Discussion

The rising incidence of LC underscores the pressing need for more effective treatment strategies. Cryoablation has emerged as a promising approach for various cancers, including non-small-cell lung cancer (NSCLC) [11,12,24]. A recent meta-analysis by Xu et al. (2023) found that cryoablation for the treatment of NSCLC was superior to radiofrequency ablation, with improved disease-free survival time, along with fewer complications and a significant reduction in recurrence rates [6]. While effective, an understanding of the MLT is necessary to assure that lung cancer destruction remains unknown. Several clinical studies have suggested an MLT in the −20 °C to −35 °C range [24,25,26]. Accordingly, in this study we utilized the A549 cell line as a model to investigate the response of lung adenocarcinoma carcinoma cells to freezing in an effort to identify the MLT. Additionally, we investigated the effect of combining cryoablation with gemcitabine, a standard chemotherapy drug as a potential adjunctive treatment path for NSCLC.

Understanding the MLT is crucial for optimizing cryoablation protocols. While previous studies have investigated MLT for various cancers, defining it for LC remains open due to the complex nature of lung tissue and varying responses to freezing. The results of this study demonstrate that A549 cells were effectively destroyed following a: (1) single freeze ≤−25 °C; (2) repeat freeze to −25 °C; (3) combination treatment with a single freeze ≤−20 °C; and (4) combination treatment with a repeat freeze ≤−15 °C (Figure 1 and Figure 3). Our reported lethal temperature of −25 °C for a repeat 5 min freeze is comparable to other published data [24] as well as other cancer types, which range from −20 °C to −40 °C [18,19,21,33,58]. Further, studies have shown that translation of the MLT as determined in in vitro studies translates well into in vivo application. For instance, Snyder et al. and others have determined that the MLT for cardiac cells was −30 °C in vitro, and that translated directly to in vivo, wherein the MLT has been shown to be in the −25 °C to −30 °C range [59,60,61]. Gage et al. and others have shown similar translation of in vitro MLT studies on prostate cancer to clinical application wherein the MLT of −40 °C has been established [18,62]. Similarly, studies have shown translation of in vitro MLT in bladder and kidney cancer to clinical MLT [26,33,63,64]. In these and other studies, what has been observed is that the in vitro MLT typically provides a conservative minimal target temperature (e.g., ~5 °C colder) compared to that found in vivo. Importantly, the in vitro data presented herein correlate well with studies on the successful application of cryoablation clinically to treat NSCLC, which, when coupled with the literature on cryoablation system performance, indirectly suggests an MLT in the −20 °C to −35 °C range [22,23,24,25,26,65,66]. Importantly, the data suggest that the application of combination treatment and repeat freezing can shift the MLT from ≤−25 °C to ≤−15 °C, thereby potentially broadening the applicability of cryoablation for NSCLC treatment (Figure 3).

The addition of gemcitabine pre-treatment increased the lethal temperature to −20 °C for a single 5 min exposure and −15 °C for a repeat exposure, representing a significant improvement in the likelihood of complete cancer destruction. Depending on the cryogen utilized, this would represent an increase in the ablative volume within a given frozen mass of 40%. For instance, following a repeat 5 min freeze (5/5/5) with 17Ga PSN cryoneedle, the ablative volume would increase from ~55% to ~75% of the frozen tissue volume (e.g., 8.58 cm^3^ (±0.32) to 12.1 cm^3^ (±0.43)) [19]. Similarly, using a standard repeat 10 min freeze procedure (10/5/10) with 17Ga argon cryoneedle, the ablative volume would increase from ~42% to ~62% of the frozen tissue volume [33,58,65,66].

While the potential increased ablative efficacy using combination treatment is promising, today’s cryosurgical technologies require a percutaneous approach, thereby complicating procedures and limiting use in treating LC [14,15,67]. Further, many protocols using argon-based CA devices utilize two to three freeze/thaw cycles with 10–15 min active freezing and 3–5 min of passive thawing [7,58,65,66,68]. The development of next-generation CA devices may assist in decreasing procedure time, given their utilization of ultra-cold cryogens which creates colder ice in shorter periods of time [19,22,23]. Further, the development of next-generation bronchoscope compatible cryocatheters could overcome this limitation, thereby enabling the increased use of CA to treat LC [69,70,71,72].

In addition to establishing the MLT for A549 cells and the potential impact of combination treatment, we also examined the mechanisms underlying the increased cell death following combination treatment. Flow cytometry analysis showed a substantial increase in necrotic cells in freeze alone and combination-treated samples (Figure 4). Protein analysis revealed that the increased cell death observed correlated with the upregulation of phosphorylated MLKL, a key mediator of necroptosis, and the downregulation of RIP3 kinase, an upstream regulator (Figure 5A–C).

Necroptosis is a fairly recently recognized form of programmed cell death (PCD). Prior to its discovery, necrosis and apoptosis were considered opposite and separate ends of the cell death spectrum [73,74,75]. Necrosis results in an unregulated swelling and bursting of cellular components, thereby stimulating an inflammatory response [74]. Conversely, the highly regulated process of apoptosis is characterized by chromatin condensation, cell shrinkage, and blebbing into apoptotic bodies that are then phagocytosed to avoid an inflammatory response [74]. Necroptosis, or programmed necrosis, involves RIP3 kinase and MLKL as primary mediators and shares characteristics of both classical apoptosis and necrosis—specifically, the lytic effect of necrosis triggered by similar biochemical signals as apoptosis [76,77]. Cisplatin, another chemotherapeutic drug used in the treatment of NSCLC, has also been shown to induce cell death in A549 cells through MLKL-mediated necroptosis [78]. Cisplatin works through a similar mechanism of action as gemcitabine, ultimately inhibiting DNA replication and repair in the nucleus, leading to cell death. In this study, we observed a significant increase in MLKL phosphorylation as well as a decrease in RIPK3 protein levels in samples treated with the combination of gemcitabine and freezing compared to freeze and gemcitabine alone, supporting the role of necroptosis-mediated cell death in combination samples (Figure 5A–C).

Analysis of Gasdermin D was also conducted to assess the potential involvement of another form of programmed necrotic cell death, pyroptosis [74]. Given that pyroptosis is primarily involved in inflammatory reactions and pathogen defense [79], we hypothesized that pyroptosis was not involved and, as such, cleaved (activated) Gasdermin D would not be detected. Proteomic analysis of gemcitabine, freeze, and combination-treated samples revealed no Gasdermin D cleavage, suggesting no involvement of pyroptosis (Figure 6A,B). Similarly, Caspase 3 activation was analyzed to further assess apoptotic involvement. Increased Caspase 3 cleavage (active) was not observed in any of the conditions; however, pro-caspase 3 levels were found to increase in combination-treated samples (Figure 7A,B). This may indicate an initiation of the apoptotic cascade followed by a transition to necroptosis. As such, further investigation is necessary to determine if different modes of cell death are active at other time points post freeze.

As advancements in CA continue to progress to more mainstream use for LC, utilization of adjuvant strategies may offer an advanced treatment option for patients. CA is known to be immunomodulatory as a monotherapy [80,81,82], and NSCLC has been shown to respond to immune activation [83,84]. Indeed, there are a growing number of studies investigating the use of immune checkpoint inhibitors (ICIs) and other targeted immunotherapy molecules in lung cancer [45,85]. Additionally, a study by Gravett et al. (2019) found that subcytotoxic concentrations of chemotherapeutic drugs, including gemcitabine, upregulated CD95/FAS death receptors and NKG2D ligands in A549 cells, suggesting a sensitization pathway for immune activation [86]. Given the side effects of chemotherapeutic drugs, the use of a sublethal dose of gemcitabine for combination treatment offers potential benefit for patients. For instance, reduced chemotherapy doses combined with the minimally invasive nature of CA may allow expansion of the patient pool eligible for treatment when applied clinically. Another issue associated with chemotherapy is the development of/selection for a chemo-resistive subpopulation of cancer cells which survive and divide over time, resulting in treatment failure [87,88]. As such, in this study we elected to allow the samples to “recover” for the extended period of 7 days (versus 3 days which is typically reported) to assess the potential for treatment-resistive subpopulations (surviving cells) to regrow in extended culture. The results demonstrate that there was no observed survival or regrowth following the combination procedure at ≤−20 °C. Taken together with the results of this study, local tumor delivery of lower doses of traditional chemotherapeutic drugs prior to CA in combination with ICIs may hold promise for future investigations of combinatorial use in NSCLC. Further, lower toxicity combination protocols coupled with shorter freeze durations (e.g., 5 min) applied in either a single or repeat freeze protocol using next-generation bronchoscope compatible cryotechnologies could further expand the use of CA to treat LC [19,30,69,70,71].

The results of this study are promising, suggesting that both cryoablation alone and in combination with gemcitabine may offer an effective treatment path for LC. While promising, there are several limitations to this study. Firstly, this study utilized only one cell line (A549 cells) as the model. The correlation of our findings and the literature are compelling and suggest this response to be universal. However, further studies with additional LC cell models are necessary to confirm this. Secondly, these studies were conducted using adherent cell monolayers. Based on the promising results, next-step studies will focus on investigations using 3-dimensional tissue engineered lung cancer tumor models and in vivo models to further validate the benefit of combinatorial treatment. Third, as with any in vitro assessment, translation to clinical application is unlikely to be linear. Further, the in vitro nature of this study provides ideal conditions for cell survival and recovery following freezing. In vivo, additional stressors including ischemia and nutrient deprivation due to vascular disruption and activation of an immune response, among others, would likely result in increased cell death [16,17,45,46]. Fourth, the concentration of gemcitabine used in combination treatment studies was based on the LC50 for A549 cells in vitro and, as such, was highly subclinical (~1/1000 clinical dose). In vivo, a typical dose is ~1000 mg/m^2^ administered over 30 min [57], which equates to ~90 µM in vitro. Initial experiments revealed that concentrations above 500 nM resulted in complete A549 cell death in vitro, and therefore, the range of 10–100 nM was selected to determine the in vitro LC50 for A549 cells. These studies resulted in the selection of 75 nM for combination treatment experiments. Regression of the data suggests a theoretical EC50 (effective concentration yielding 50% cell death) for the combination treatment used when combining a −10 °C repeat freeze and 75 nM of gemcitabine or a −20 °C repeat freeze and 25 nM of gemcitabine. Further, in this study we exposed A549 cells directly to gemcitabine, whereas in vivo exposure is typically via vascular delivery. While direct application is more effective than vascular delivery, the 75 nM dose used in this study was ~1/1000 that of a clinical dose equivalent (90 µM) and, as such, is still substantially lower than what LC cells are exposed to in vivo. Drug delivery to a tumor via direct injection or nanoparticles may allow for a greater concentration to reach the targeted tissue while minimizing systemic effects. However, there are some challenges related to dosing and distribution which remain [89,90,91]. Fifth, while gemcitabine has been used extensively to treat NSCLC, it is often used in combination with other drugs to treat lung adenocarcinoma clinically to increase efficacy. While less common as a primary drug clinically today, our results suggest that the use of gemcitabine in combination with freezing has a similar, potentially improved, effect as when combined with other drugs [55,92,93,94]. Sixth, combination studies were limited to evaluating a single chemotherapeutic agent, gemcitabine, in an effort to establish evidence of the benefit combinatorial cryo/chemo treatment for LC. Based on the promising results presented herein, studies with other drugs, combinations, doses, etc., are now warranted. Lastly, we recognize that due to the limited time points analyzed and in vitro nature of the study, one cannot rule out apoptosis and pyroptosis playing a role in cell death following either freezing or combination treatment in vivo. While these cannot be ruled out, the data do strongly support the role of necroptosis in cell death following both freezing and combination treatment. We recognize that including inhibitors of necroptosis would lend further support to our hypothesis and, as such, future studies will utilize these agents.

## 5. Conclusions

In conclusion, this study demonstrates the potential of cryoablation alone or in combination with gemcitabine as a promising treatment strategy for LC. The results demonstrate that repeat freezing to −25 °C effectively destroys NSCLC A549 cells, whereas the combination of gemcitabine pre-treatment enhances A549 cell susceptibility to freezing, thereby yielding complete cell death following either a single freeze to −20 °C or a repeat freeze to −15 °C. These results suggest that the synergy between cryoablation and chemotherapy may offer enhanced efficacy in targeting NSCLC, thereby potentially expanding the therapeutic window. Mechanistic investigations revealed a predominant role of necroptosis in cell death following combination treatment. These findings contribute to the ongoing efforts to improve treatment options for NSCLC, offering a promising approach that warrants further investigation in preclinical and clinical settings. By elucidating the optimal cryoablation parameters and identifying potential synergies with chemotherapy, this study lays the groundwork for future studies aimed at enhancing the efficacy of cryoablation for NSCLC treatment. While further research is warranted to address remaining questions and optimize clinical translation, our findings underscore the therapeutic potential of cryoablation alone or in combination with chemotherapy for the management of NSCLC.

## Figures and Tables

**Figure 1 biomedicines-12-01239-f001:**
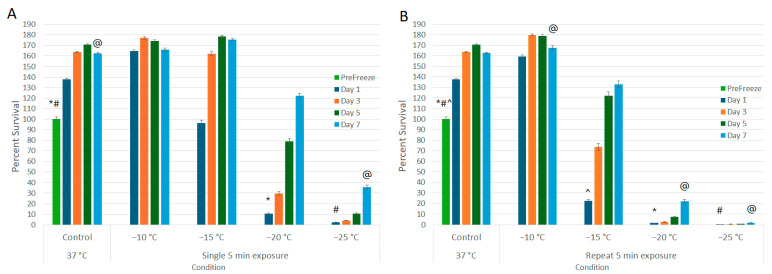
Assessment of Lung Cancer Cell Viability and Recovery Following a Single (**A**) or Repeat (**B**) Freeze Protocol. A549 cells were subjected to a (**A**) single or (**B**) repeat 5 min freeze at −10, −15, −20, and −25 °C, and survival was assessed over seven days post treatment. Data suggest that complete cell death with no recovery is attained following a repeat freeze to −25 °C, whereas a single freeze results in a substantial level of cell death followed by recovery in culture. Similarly, a repeat freeze at −20 °C yielded near-complete cell death; however, cell recovery was noted over the 7-day assessment interval. (*,#,@,^ = *p* < 0.01, respectively).

**Figure 2 biomedicines-12-01239-f002:**
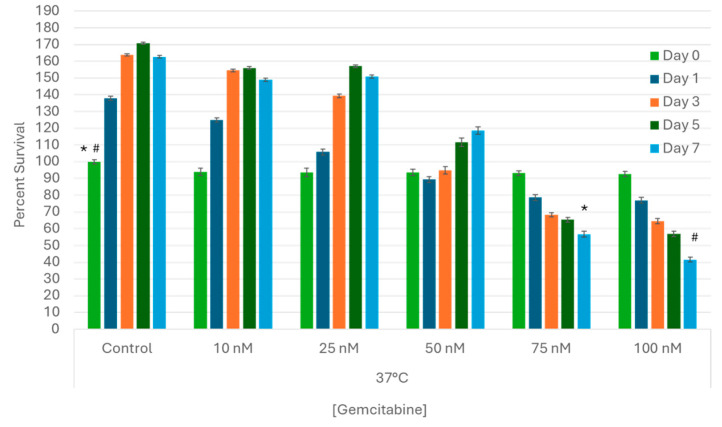
Assessment of Gemcitabine Treatment on Lung Cancer Cell Survival. A549 cells were subjected to 24 h pretreatment with subclinical doses (10, 25, 50, 75, or 100 nM) of gemcitabine at 37 °C. Cell viability declined steadily in samples exposed to 75 and 100 nM over seven days post treatment, but complete destruction was not observed. Based on the dose–response studies, a gemcitabine concentration of 75 nM was determined to be the LC50 for A549 cells in vitro. (*,# = *p* < 0.01, respectively).

**Figure 3 biomedicines-12-01239-f003:**
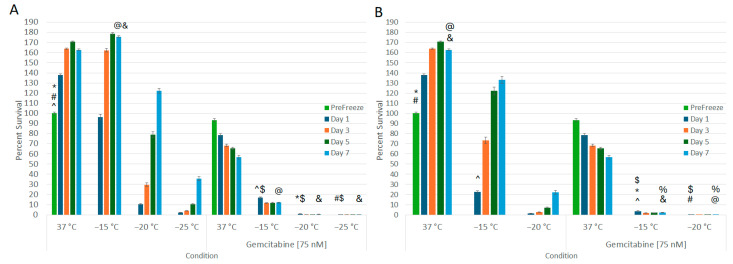
Effect of Adjunctive Gemcitabine Pretreatment in Combination with Freezing on Lung Cancer Cell Survival. A549 cells were subjected to 24 h pretreatment with 75 nM of gemcitabine followed by a single (**A**) or repeat (**B**) 5 min freeze at −15, −20, and −25 °C. Data suggest that the combination of gemcitabine pretreatment and single freeze to −20 °C (**A**) or repeat freeze to −15 °C (**B**) results in complete lung cancer cell death. This was found to be significantly different from freeze alone or gemcitabine alone treatments. (*,#,@,&,^ = *p* < 0.01, respectively. $,% *p* = NS).

**Figure 4 biomedicines-12-01239-f004:**
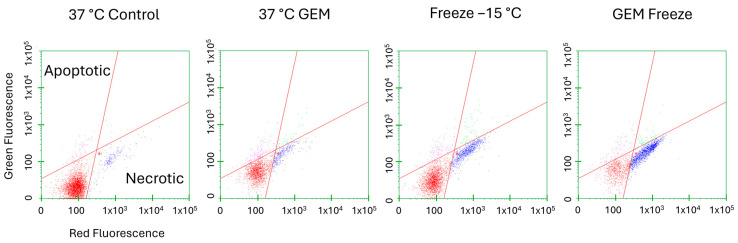
Assessment of Cell Death Subpopulations Following Freezing or Combination Treatments. Living (lower left, red dots), apoptotic (upper left, pink dots), and necrotic (lower right, blue dots) subpopulations were analyzed as a percentage of the whole sample. A549 cells were subjected to −15 °C alone or in combination with 75 nM of gemcitabine and assessed using microfluidic flow cytometry 24 h post treatment. Data illustrate necrosis as the predominant mode of cell death, with apoptosis contributing to a lesser degree following freezing. When combination treatment was applied, apoptosis levels diminished and necrosis levels increased further. (X-axis = log red fluorescent intensity, Y-axis = log green fluorescent intensity).

**Figure 5 biomedicines-12-01239-f005:**
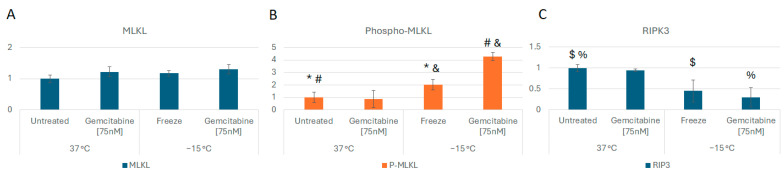
Analysis of Necroptosis Cell Death Protein Levels Following Freezing or Combination Treatments. A549 cells were subjected to −15 °C alone or in combination with 75 nM of gemcitabine, protein isolated 6 h post treatment and assessed using the WES Simple Western system. (**A**) analysis of MLKL levels revealed no change between controls and treated samples. (**B**) Analysis of phospho-MLKL levels revealed a 2- and 4-fold increase in the −15 °C and combination samples, respectively, over controls, indicating the activation of necroptosis. (**C**) RIPK protein levels were found to decrease in freeze and combination samples, further supporting necroptosis involvement. (* *p* = 0.04; $ *p* = 0.03; % *p* < 0.05; #,& *p* < 0.01).

**Figure 6 biomedicines-12-01239-f006:**
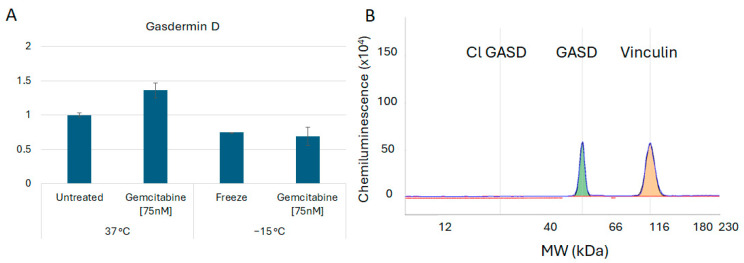
Analysis of Gasdermin B Cell Death Protein Levels Following Freezing or Combination Treatments. A549 cells were subjected to −15 °C alone or in combination with 75 nM of gemcitabine, protein isolated 6 h post treatment and assessed using the WES Simple Western system. (**A**) Analysis of Gasdermin D levels revealed no significant changes nor (**B**) any cleavage products following any of the treatments, suggesting that pyroptosis is not involved in vitro. (**B**) is a representative electropherogram from the combination gemcitabine and freezing treatment showing the protein of interest, Gasdermin D, multiplexed with the normalization protein, vinculin.

**Figure 7 biomedicines-12-01239-f007:**
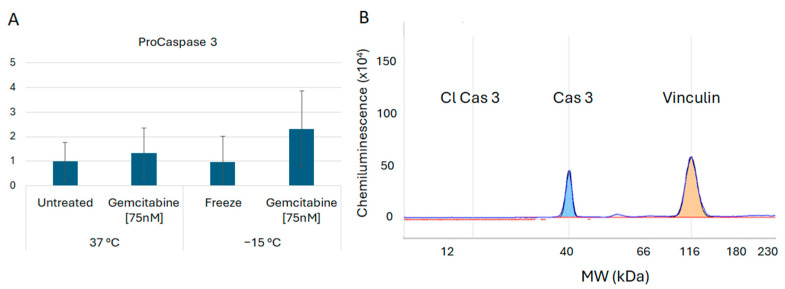
Analysis of Caspase 3 Cell Death Protein Levels Following Freezing or Combination Treatments. A549 cells were subjected to −15 °C alone or in combination with 75 nM of gemcitabine, protein isolated 6 h post treatment and assessed using the WES Simple Western system. (**A**) Analysis of Caspase 3 revealed an increase in pro-Caspase 3 levels post treatment; however, (**B**) no cleavage products (active) were detected. This suggests that the apoptotic cascade may be activated further downstream as a secondary form of cell death. (**B**) is a representative electropherogram from the combination gemcitabine and freezing treatment showing the protein of interest, caspase 3, multiplexed with the normalization protein, vinculin.

**Table 1 biomedicines-12-01239-t001:** Quantification of the Mode of Cell Death Associated with Freezing and Combination Treatments.

Condition	% Live	% Apoptotic	% Necrotic
37 °C Control	89.7 (4.9)	2.1 (1.6)	7.9 (5.2)
37 °C GEM 75nM	71.8 (6.1)	6.9 (5.4)	19.9 (7.3)
Freeze (−15 °C)	64.1 (6.6)	4.0 (2.9)	30.6 (8.4)
GEM Freeze	27.7 (2.5)	2.9 (1.0)	67.7 (3.8)

## Data Availability

The data that support the findings of this study are available from CPSI Biotech, but restrictions apply to the availability of these data, which were used under license for the current study, and so are not publicly available. Data are, however, available from the authors upon reasonable request and with the permission of CPSI Biotech.

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
