# Peer review of "Investigation of Lung Cancer Cell Response to Cryoablation and Adjunctive Gemcitabine-Based Cryo-Chemotherapy Using the A549 Cell Line"

_biomedicines, 2024, doi:10.3390/biomedicines12061239_

Round 1

Reviewer 1 Report

Comments and Suggestions for Authors

The authors of this paper propose to study the effects of cryoablation, alone or in combination with gemcitabine, a chemotherapeutic agent, in the treatment of lung cancer, using an in vitro model. It is an interesting study that provides promising results and deserves to be published. The context is clearly stated and the research design is appropriate. The results are clearly presented and support the conclusions drawn by the authors.

I have only minor comments:

-        The authors state that one aim of their study was to determine the MLT of NSCLC using A549 cells. To what extent is this MLT determined in vitro representative of an in vivo MLT? As they explain later in the discussion (and I particularly appreciated this part), the in vitro/in vivo correlation could be challenging.

-        Similarly, the authors used a (very common) cell line, but there are other NSCLC cell lines. Have the authors performed other experiments with cell lines other than A549 that could strengthen their results and allow to generalize their conclusion? Or do they know of any published data in this regard?

I have also noticed the following typos:

L 97: 3 and 2 should be superscripted: 103 and cm2

L99: Gemcitabine and chemotherapy. For the latter, I am not sure if it should appear here as this paragraph rather describes the exposure to the drug, while the freezing procedure is described afterwards.

L107: what is the meaning of nadir?

L143: protein

L149: 2µg/µl followed by two commas

L171: Instead of at the end of the sentence, (Figure 1A) should be after “single” as it only refers to this exposure condition.

The legend of Figure 1 is incomplete and should be clarified: A and B should be added in the title. The explanation should also be completed, as the protocol only refers to the single exposure experiments, while the conclusion refers to the repeated exposure.

L302: pyroptosis

Author Response

The authors of this paper propose to study the effects of cryoablation, alone or in combination with gemcitabine, a chemotherapeutic agent, in the treatment of lung cancer, using an in vitro model. It is an interesting study that provides promising results and deserves to be published. The context is clearly stated and the research design is appropriate. The results are clearly presented and support the conclusions drawn by the authors. I have only minor comments:

Comment 1: The authors state that one aim of their study was to determine the MLT of NSCLC using A549 cells. To what extent is this MLT determined in vitro representative of an in vivo MLT? As they explain later in the discussion (and I particularly appreciated this part), the in vitro/in vivo correlation could be challenging.

Response: This is an important topic.  Studies have shown that translation of the MLT as determined in in vitro studies translates well into in vivo application.  For instance, Snyder et al and others have determined that the MLT for cardiac cells was -30°C in vitro and that translated directly to in vivo wherein the MLT has been shown to be in the -25°C to -30°C range. Gage et al and others have shown similar translation of in vitro MLT studies on prostate cancer to clinical application wherein the MLT of -40°C has been established.  Similarly, studies have shown translation of in vitro MLT in bladder and kidney cancer to clinical MLT.  In these and other studies, what has been observed is that the in vitro MLT typically provides a conservative minimal target temperature (eg ~5°C colder) when applied in vivo. Expanded discussion has been added to the revised manuscript on this subject.

Comment 2:  Similarly, the authors used a (very common) cell line, but there are other NSCLC cell lines. Have the authors performed other experiments with cell lines other than A549 that could strengthen their results and allow to generalize their conclusion? Or do they know of any published data in this regard?

Response: The reviewer is correct, the A549 cell line is a very common cell used in studies.  As such we selected the A549 cell line as the model for this study.  We did not utilize other lung cancer cell lines. Reports on the successful application of cryoablation clinically to treat NSCLC coupled with knowledge of various cryoablation system performance, interpretation of the data in the literature suggests a MLT in the -20°C to -35°C range.  With that in mind we elected to utilize the A549 cell line as a model to conduct targeted investigations.  Discussion has been added to the revised manuscript on this subject.  Additionally, to further address this comment we have revised the title to add “… using the A549 cell line” to reflect the model and study limitation.

Comment 3: L 97: 3 and 2 should be superscripted: 103 and cm2. 

Response: Corrected

Comment 4: L99: Gemcitabine and chemotherapy. For the latter, I am not sure if it should appear here as this paragraph rather describes the exposure to the drug, while the freezing procedure is described afterwards.

Response: Corrected

Comment 5: L107: what is the meaning of nadir?

Response: Low point.  Clarified in test.

Comment 6: L143: protein

Response: Corrected

Comment 7: L149: 2µg/µl followed by two commas

Response: Corrected

Comment 8: L171: Instead of at the end of the sentence, (Figure 1A) should be after “single” as it only refers to this exposure condition.

Response: Corrected

Comment 9: The legend of Figure 1 is incomplete and should be clarified: A and B should be added in the title. The explanation should also be completed, as the protocol only refers to the single exposure experiments, while the conclusion refers to the repeated exposure.

Response: Corrected

Comment 10: L302: pyroptosis

Response: Corrected

Reviewer 2 Report

Comments and Suggestions for Authors

Congratulations to you on your brilliant experiment of treatment of cancer by cryoablation combined with gemcitabine. 

The entire manuscript was well written with good language and fine layout. Also there is novelty on the treatment which combine chemotherapy and cryotherapy. 

This article provides a new idea and I think it can be implemented clinically.

It will be good if after the pretreatment with gemcitabine the freezing temperature can lowered to -15 to -20 celsius, which causes less surrounding tissue damage.

I hope the procedure will be IRB approved and implemented in patients in the near future.

Author Response

Reviewer 2

Comments: Congratulations to you on your brilliant experiment of treatment of cancer by cryoablation combined with gemcitabine. The entire manuscript was well written with good language and fine layout. Also there is novelty on the treatment which combine chemotherapy and cryotherapy. This article provides a new idea and I think it can be implemented clinically. It will be good if after the pretreatment with gemcitabine the freezing temperature can lowered to -15 to -20 celsius, which causes less surrounding tissue damage. I hope the procedure will be IRB approved and implemented in patients in the near future.

Response: We thank the reviewer for their comments. We agree and are hopeful to initiate IRB studies in the near future. We also agree the reduction of damage to surrounding tissue is a very important potential benefit of the combination treatment. As risk of pneumothorax is a real concern with any ablation technique used in lung cancer, reducing the amount of non-targeted tissue damage while assuring tumor destruction should significantly reduce the risk of pneumothorax.

Reviewer 3 Report

Comments and Suggestions for Authors

The article is devoted to the study of the possibility of using cryochemotherapy to inhibit the growth of non-small cell lung carcinoma cells. The authors determined the temperature at which freezing and chemotherapy kill cells in the monolayer. Possible mechanisms of cell death during cryochemotherapy are touched upon.

The article is well written and easy to read

The introduction reflects the research problem quite well

Methods need addition

The results are described normally, I have questions about statistical data analysis

The discussion places the research results in the context of other works quite well

My questions:

1. You are studying cryochemotherapy in a monolayer of cells, but the use of 3D tumor models (at least multicellular spheroids) would significantly increase the significance of your results. I suggest repeating at least the viability analysis on 3D models. To do this, you can use regular alamar, or dissociate the spheroid and analyze cell viability after staining with sytox or propidium iodide.

2. What is the likelihood of aggressive resistant cells appearing after cryochemotherapy? Discuss. Ideally, see how cells grow after cryochemotherapy. For example, the experiment in Figure 3B is “-15C and -20C”. Let these points live for a couple more days, if the cells grow at -15, but do not grow at -20 (which obviously will happen, since there are no viable cells there), then you can justify the choice of temperature also by the lack of resistant survivors.

3. I have questions about statistical data processing:

- what tests did you use? I would suggest ANOVA/Kruskal-Wallis with a post hoc test. Describe statistical tests in methods.

-indicate the difference between data groups in all figures * - p < 0.05, ** - p < 0.01, etc.

- I think your p values are too large for such strong effects, please check for all data.

- show individual points on all graphs.

- Figure 3 – what are the conditions at the bottom of the panels?

4. Try to slightly reinterpret the data you receive:

- for 3.2. part – I believe it is better to use the term LC50 – lethal concentration, not a "dose" of gemcitabine.

- try to perform a regression of the data in Fig. 2, calculate the EC50 of gemcitabine with cryo in your case and calculate the analogue of the “therapeutic index”. How much higher or lower is your EC50 than gemcitabine itself (sub-micrograms per ml)?

5. Flow cytometry:

- show the gating strategy, how many cells were in the final gate?

- try applying compensation. I don't think this will work since the dye emissions are well resolved. But then the question arises - why do you consider the blue population to be only sytox+, and not cellevent+ sytox+?

- you do not have activation of caspase 3, and according to your interpretation caspases 3 and 7 are not activated as well (FC confirms the immunoassay data for casp 3). I would put Fig 7 with caspase 3 after FC, so you first cross out apoptosis (PS and immunoassay), then you can put Fig 6 with gasdemin D to exclude ferroptosis, and then you can confirm necroptosis with data from Fig 5. My sequence of presentation is optional, but think about it.

- Show the data from Table 1 in the form of columns that show an increase in “necrotic” cells, I think this will be more clear.

Little things:

- Figure 6 and 7, increase font size of the y-axis label in panels B

- add doi for all links (no dois in 25, 27, 31)

- day 7 or D7 – make the designations the same.

Overall, the article is interesting, but needs revision.

Author Response

Reviewer 3

The article is devoted to the study of the possibility of using cryochemotherapy to inhibit the growth of non-small cell lung carcinoma cells. The authors determined the temperature at which freezing and chemotherapy kill cells in the monolayer. Possible mechanisms of cell death during cryochemotherapy are touched upon.

General Comments:

  • The article is well written and easy to read
  • The introduction reflects the research problem quite well
  • Methods need addition
  • The results are described normally, I have questions about statistical data analysis
  • The discussion places the research results in the context of other works quite well

Response: Thank you for the comments.  We have addressed the methods in the revised manuscript (detailed below).

Specific Comments:

Comment 1:  You are studying cryochemotherapy in a monolayer of cells, but the use of 3D tumor models (at least multicellular spheroids) would significantly increase the significance of your results. I suggest repeating at least the viability analysis on 3D models. To do this, you can use regular alamar, or dissociate the spheroid and analyze cell viability after staining with sytox or propidium iodide.

Response:  Thank you for the comment.  We agree 3D models (and in vivo) studies will further this line of investigation. Our next step is to conduct combination studies in a 3D tissue engineered lung cancer model in association with a new cryoablation device and protocols.  We have added discussion of this in the revised manuscript.

Comment 2: What is the likelihood of aggressive resistant cells appearing after cryochemotherapy? Discuss. Ideally, see how cells grow after cryochemotherapy. For example, the experiment in Figure 3B is “-15C and -20C”. Let these points live for a couple more days, if the cells grow at -15, but do not grow at -20 (which obviously will happen, since there are no viable cells there), then you can justify the choice of temperature also by the lack of resistant survivors.

Response: This is an excellent point and one which we made but downplayed due to the in vitro nature of the model in the original manuscript. In the study we selected the 7 day recovery interval to do just this. It is well established in the literature that there can be a subpopulation of cells which are chemo resistant in vivo. Additionally chemoresistance can develop overtime.  It is also well established that resistance to temperatures below the MLT does not occur in vitro or in vivo.  It is this reason wherein the combination of chemotherapy and cryoablation offers potential benefit to overcome and destroy chemo resistant cancer cells.  This has been shown to be effective clinically for the treatment of prostate cancer but has yet to be explored for lung cancer. In this study we elected to allow the samples to “recover” for the extended period of 7 days (versus 3 days which is typically reposted) to show precisely the point that there are no survivors or cell regrowth following the combination procedure at ≤ -20°C.  We have added discussion on this topic to the revised manuscript.

Comment 3: I have questions about statistical data processing:

- what tests did you use? I would suggest ANOVA/Kruskal-Wallis with a post hoc test. Describe statistical tests in methods.

Response: We apologize for neglecting to describe the statistical method in the methods section.  We thank the reviewer for pointing this out.  We did use ANOVA for statistical analysis.  This has been added in the methods section.

Comment 4: indicate the difference between data groups in all figures * - p < 0.05, ** - p < 0.01, etc.

Response: Revised

Comment 5:  I think your p values are too large for such strong effects, please check for all data.

Response: the data has been checked and confirmed.

Comment 6: show individual points on all graphs.

Response: All figures have been revised to indicate points of discussion.

Comment 7:  Figure 3 – what are the conditions at the bottom of the panels?

Response: Freeze exposure temperature with and without Gemcitabine pretreatment. Clarified in Figure Legend

Comment 8:  Try to slightly reinterpret the data you receive:

- for 3.2. part – I believe it is better to use the term LC50 – lethal concentration, not a "dose" of gemcitabine.

Response: Revised to lethal concentration

Comment 9: try to perform a regression of the data in Fig. 2, calculate the EC50 of gemcitabine with cryo in your case and calculate the analogue of the “therapeutic index”. How much higher or lower is your EC50 than gemcitabine itself (sub-micrograms per ml)?

Response: The lethal concentration of gemcitabine alone in vitro for A549 cells is 500nM to obtain 100% cell death. Dose response studies determined the EC50 was 75nM. When the combination of freezing to -20°C and 75nM was applied 100% cell death was attained. We did not conduct studies with lower concentrations of gemcitabine and freezing, however, the data does suggest the EC50 of the combination treatment of freezing and 75nM gemcitabine is around -10°C versus -25°C for freezing alone.

Comment 10:  Flow cytometry:

- show the gating strategy, how many cells were in the final gate?

- try applying compensation. I don't think this will work since the dye emissions are well resolved. But then the question arises - why do you consider the blue population to be only sytox+, and not cellevent+ sytox+?

Response: We appreciate these two comments and we have revised the text of the methods section to clarify the procedure. Briefly, 5,000 events per sample were acquired and between 3500 and 4650 cells were within the final gates per experiment after debris and doublets were excluded. Compensation techniques were tested but found to be unnecessary in the resolution of these specific fluorochromes. The population of cells shifted to the right is positive for sytox red and depicted blue in the image. If cells were positive for both fluorochromes we would expect them to appear midway between each population, which we have seen in other fluorochrome combinations, but not here.

Comment 11:  you do not have activation of caspase 3, and according to your interpretation caspases 3 and 7 are not activated as well (FC confirms the immunoassay data for casp 3). I would put Fig 7 with caspase 3 after FC, so you first cross out apoptosis (PS and immunoassay), then you can put Fig 6 with gasdemin D to exclude ferroptosis, and then you can confirm necroptosis with data from Fig 5. My sequence of presentation is optional, but think about it.

Response: Thank you for the suggestion.  We considered this however elected to group the data presentation based on analysis type for clarity. 

Comment 12: Show the data from Table 1 in the form of columns that show an increase in “necrotic” cells, I think this will be more clear.

Response: thank you for the suggestion.  We have reordered the table placing necrosis as the last column with conditions arranged in ascending order (top to bottom) from least (control) to greatest (combination) necrosis.

Comment 13: Figure 6 and 7, increase font size of the y-axis label in panels B

Response: Revised

Comment 14:  add doi for all links (no dois in 25, 27, 31)

Response: Revised

Comment 15:  day 7 or D7 – make the designations the same.

Response: Revised

Comment 15: Overall, the article is interesting, but needs revision.

Response: Thank you for your valuable comments.

Round 2

Reviewer 3 Report

Comments and Suggestions for Authors

The authors answered some of my comments properly.

They did not provide additional experiments but explained it in discussion section.

My major concern about statistics used remaines unanswered:

- What post hoc test you used after One-way ANOVA?

- Try to show individual points in a figures.

- It would be nice if you include your statistics analysis as a supplementary file.

I can recommend the article for publication after minor revision.

Author Response

Reviewer 3:

Comment: They did not provide additional experiments but explained it in discussion section.

Response: Correct. In addition to explaining in the discussion we also revised the title of the manuscript for clarity.  We have added text to the introduction and abstract as well to provide further clarity.

Comment: What post hoc test you used after One-way ANOVA?

Response: The post hoc analysis was Tukey HDS.  The text has been revised appropriately. Specifically, analysis was performed using the “One-way ANOVA (Analysis Of Variance) with post-hoc Tukey HSD (Honestly Significant Difference) Test Calculator for comparing multiple treatments” statistical analysis program (https://astatsa.com/OneWay_Anova_with_TukeyHSD)

Comment: Try to show individual points in a figures.

Response: Comparative points are now highlighted in the Figure captions.  Individual points of comparison are also noted in the figures with symbols.

Comment: It would be nice if you include your statistics analysis as a supplementary file.

Response:  We appreciate the comment.  As we have utilized standard analysis and have indicated the statistical software utilized, we feel it is unnecessary to add a supplementary file.

Comment:  I can recommend the article for publication after minor revision.

Response: We thank the reviewer for their diligence and thorough comments.  The comments provided by all the reviewers and subsequent revisions have resulted in a much clearer, focused manuscript for publication.